# Automatic Bias Control Technique of Dual-Parallel Mach–Zehnder Modulator Based on Simulated Annealing Algorithm for Quadrupled Signal Generation

Youngseok Bae *, Sunghoon Jang, Sungjun Yoo, Minwoo Yi, Joonhyung Ryoo and Jinwoo Shin

Quantum Physics Technology Directorate, Agency for Defense Development, Daejeon 34186, Korea; jangsh@add.re.kr (S.J.); sj_ryu@add.re.kr (S.Y.); minuyi@add.re.kr (M.Y.); yoojh@add.re.kr (J.R.); sjinu@add.re.kr (J.S.)
* Correspondence: youngseok.bae@add.re.kr; Tel.: +82-42-821-0498

**Abstract:** The radio frequency (RF) signal generation method using an external modulator is widely used in microwave photonics applications because it has the advantage of being able to generate coherent and stable RF signals with a higher resolution performance compared to the conventional method. A Mach–Zehnder modulator is widely used as an external modulator due to its high electro-optic coefficients and low attenuation characteristics but has a critical problem in that its electrical characteristics are changed by external environments such as temperature. In this paper, we considered the stabilization configuration to overcome this problem and propose an automatic bias control technique based on the simulated annealing algorithm of a dual-parallel Mach–Zehnder modulator (DPMZM) for quadruple signal generation. The proposed technique searches for the bias voltages of the modulator in real-time through the temperature test. In addition, the output of the quadrupled signal of the DPMZM is constantly controlled throughout the temperature range. Finally, it is confirmed that signals of a 10 GHz and 22 GHz frequency are generated using the intermediate frequency signals of a 2.5 GHz and 5.5 GHz frequency with the proposed automatic bias control technique, respectively.

**Keywords:** automatic bias control; dual-parallel mach–zehnder modulator; simulated annealing algorithm; quadrupled signal generation

## 1. Introduction

The microwave photonics technology is an alternative to overcome the technical limitations of existing electronic-based radio frequency (RF) signal generation, detection, and processing [1,2]. There are various methods of the photonics-based RF signal generation method, among which the method using an external modulator is widely used because it is relatively simple to implement and can generate a coherent signal [3]. Among the modulators, a Mach–Zehnder modulator (MZM) is widely used because of its low-driving voltage, large modulation bandwidth, and low signal chirp [4,5]. An MZM is made of indium phosphide (InP), lithium niobate (LiNbO₃), or gallium arsenide (GaAs) materials. The refractive index is modified when the electric field applied, and the nonlinear effect derives from the optical interference [6].

Due to the inherent nonlinear characteristics of the modulator, the optimal bias voltages scanning and stabilization techniques of the modulator are essential [7–9]. For example, the various techniques for maintaining the bias of an MZM for each of the applications have been developed [10–16]. For example, The previous researches in [10] and in [11] proposed an asymmetric dithering signal and an output signal feedback loop method for an optical quadrature phase shift keying (QPSK) modulation, respectively. In [12], in-phase and quadrature modulation scheme using two dither signals for optical QPSK modulation was proposed. The previous study in [13] monitored the modulated average output to

control the bias voltage, and a pilot tone-based method for an electro-optic analog-to-digital converter was proposed in [14]. In [15], the output power was fed back for bias locking. In [16], a method of increasing the dither detection sensitivity using a three-by-three optical coupler was proposed. However, most of the previous studies have been performed for optical QPSK modulation [10–12]. For optical QPSK modulation using a dual-parallel Mach–Zehnder modulator (DPMZM), the phase bias of the DPMZM (MZM-m in Figure 1) should be set to be the right quadrature transmission point ($+\pi/2$) or the left quadrature transmission point ($-\pi/2$). On the contrary, microwave photonics technology employs the techniques such as double, quadrupled, or octupling signal generation using the applied intermediate frequency (IF) signal of the modulator for RF signal generation [17–20]. Since the phase bias of the DPMZM should be set to be the minimum transmission point (MITP) of the maximum transmission point (MATP) in order to generate the quadrupled signal. In addition, the bias condition of the phase bias of the DPMZM depends on the bias conditions of an in- and quadrature-phase bias of the DPMZM (MZM-I and MZM-Q in Figure 1) [21]. Therefore, a different bias search technique is required. However, the frequency up-conversion method using an external modulator has a disadvantage in that the structure of the bias controller is complex [16].

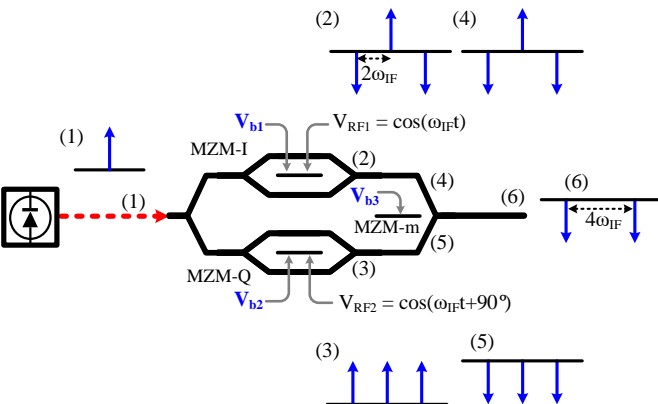

**Figure 1.** Structure of the DPMZM and principle of the quadrupled signal generation.

In this paper, we propose an automatic bias control technique for a dual-parallel MZM (DPMZM) based on a simulated annealing algorithm for generating quadrupled signals for use in frequency up-conversion applications such as photonic radar. The organization of this paper is as follows. Section 2 describes a control method based on simulated annealing algorithm proposed for automatic bias control technique. In Section 3, the proposed technique is verified through a temperature test, and the generated signals according to the various intermediate frequency (IF) signals are shown. The test results of the proposed technique is discussed in Section 4. Finally, Section 5 shows the conclusion.

## 2. The Proposed Automatic Bias Control Technique

Figure 1 shows the structure of the DPMZM which has two sub-MZMs (MZM-I and MZM-Q) on both arms of the main MZM (MZM-m) and the principle of the quadrupled signal generation. In Figure 1, $V_{b1}$, $V_{b2}$, and $V_{b3}$ are the bias voltages of two sub-MZMs and the main MZM of the DPMZM, respectively.

When the phase difference of the RF signal applied to the two sub-MZMs is $\pi/2$ and the two sub-MZMs and the main MZM operate at the MATP and the MITP, respectively, the output signal of the DPMZM can be expressed as [21–23]:

$$
\begin{aligned}
E_{\text{DPMZM}}(t) &= \frac{1}{2} E_{\text{c}} \cos(\omega_c t) \left[ \cos\{m\cos(\omega_{\text{IF}} t)\} - \cos\{m\cos(\omega_{\text{IF}} t + \pi/2)\} \right] \\
&= \frac{1}{2} E_{\text{c}} \cos(\omega_c t) \left[ J_0(m) + 2\sum_{n=1}^{\infty} (-1)^n J_{2n}(m) \cos(2n\omega_{\text{IF}} t) \right. \\
&\qquad\qquad \left. - J_0(m) + 2\sum_{n=1}^{\infty} J_{2n}(m) \cos(2n\omega_{\text{IF}} t) \right] \\
&\approx -E_{\text{c}} J_2(m) \left[ \cos\{(\omega_c + 2\omega_{\text{IF}})t\} + \cos\{(\omega_c - 2\omega_{\text{IF}})t\} \right]
\end{aligned}
\tag{1}
$$

where $E_{\text{c}}$ and $\omega_{\text{c}}$ are the amplitude and angular frequency of the optical source, $m$ is the modulation index of the two sub-MZMs, $\omega_{\text{IF}}$ is the angular frequency of the applied signal, and $J_n$ means the $n$th-order Bessel function of the first kind. Algorithm 1 shows the proposed automatic bias control technique of the DPMZM based on the simulated annealing algorithm for quadrupled signal generation. The proposed technique consists of two parts: The rough bias points scanning part with a large voltage step and the precise bias points searching part with the dense voltage step.

---

**Algorithm 1:** Bias points searching algorithm.

---

    **Input:** Output signal intensity of DPMZM
    **Output:** Bias voltages of DPMZM, $V_{\text{b1}}, V_{\text{b2}}, V_{\text{b3}}$
    **Step:** Rough bias points search with large step
1  **begin**
2     **for** $V_{\text{b3}} \leftarrow -V_\pi$ **to** $V_\pi/2$ **by** $V_\pi/2$ **do**
3         **for** $V_{\text{b1}} \leftarrow -V_\pi$ **to** $V_\pi$ **by** $V_{\text{inc}}$ **do**
4             **for** $V_{\text{b2}} \leftarrow -V_\pi$ **to** $V_\pi$ **by** $V_{\text{inc}}$ **do**
5                 $f(V_{\text{b1}}, V_{\text{b2}}, V_{\text{b3}}) \rightarrow \text{tmp}[V_{\text{b1}}][V_{\text{b2}}]$;
6             **end**
7         **end**
8         $\text{tmp}[V_{\text{b1}}][V_{\text{b2}}]|_{V_{\text{b3}}} \leftarrow \max \text{tmp}[V_{\text{b1}}][V_{\text{b2}}]$;
9     **end**
10    $(V'_{\text{b1}}, V'_{\text{b2}}) \leftarrow \underset{V_{\text{b1}}, V_{\text{b2}}}{\arg\max} \, \text{tmp}[V_{\text{b1}}][V_{\text{b2}}]|_{V_{\text{b3}}}$;
11    **for** $V_{\text{b3}} \leftarrow -V_\pi$ **to** $V_\pi$ **by** $V_{\text{inc}}$ **do**
12       $V'_{\text{b3}} \leftarrow \underset{V_{\text{b3}}}{\arg\min} \, f(V'_{\text{b1}}, V'_{\text{b2}}, V_{\text{b3}})$;
13    **end**
14 **end**
    **Step:** Precise bias points search with dense step
15 **begin**
16    **for** $V_{\text{b1}} \leftarrow -V_\pi$ **to** $V_\pi$ **by** $V'_{\text{inc}}$ **do**
17       **for** $V_{\text{b2}} \leftarrow -V_\pi$ **to** $V_\pi$ **by** $V'_{\text{inc}}$ **do**
18          $(V_{\text{b1}}^{\max}, V_{\text{b2}}^{\max}) \leftarrow \underset{V_{\text{b1}}, V_{\text{b2}}}{\arg\max} \, f(V_{\text{b1}}, V_{\text{b2}}, V'_{\text{b3}})$;
19       **end**
20    **end**
21    **for** $V_{\text{b3}} \leftarrow -V_\pi$ **to** $V_\pi$ **by** $V'_{\text{inc}}$ **do**
22       $V_{\text{b3}}^{\min} \leftarrow \underset{V_{\text{b3}}}{\arg\min} \, f(V_{\text{b1}}^{\max}, V_{\text{b2}}^{\max}, V_{\text{b3}})$;
23    **end**
24    $\{V_{\text{b1}}, V_{\text{b2}}, V_{\text{b3}}\} \leftarrow \{V_{\text{b1}}^{\max}, V_{\text{b2}}^{\max}, V_{\text{b3}}^{\min}\}$;
25 **end**

---

In the algorithm, $f(V_{\text{b1}}, V_{\text{b2}}, V_{\text{b3}})$ means the output optical signal intensity of the DPMZM when the bias voltages $V_{\text{b1}}$, $V_{\text{b2}}$, and $V_{\text{b3}}$ are applied to the sub-MZMs and the main MZM of the DPMZM, respectively. In order to find the bias point of DPMZM roughly in the first step, $V_{\text{b3}}$ alters from $-V_\pi$ to $V_\pi/2$ with four values. In each bias condition of

$V_{b3}$, $V_{b1}$ and $V_{b2}$ are changed in the $2V_\pi$ range with the voltage step $V_{inc}$, respectively. $V'_{b1}$ and $V'_{b2}$ are searched bias voltages of the two-sub-MZM that maximize the output signal intensity of the DPMZM among each $V_{b3}$ conditions. In the $V'_{b1}$ and $V'_{b2}$ condition, $V_{b3}$ is changed in the $2V_\pi$ range with the voltage step $V_{inc}$. $V'_{b3}$ is found the bias voltage of the main MZM that minimize the output signal intensity. The voltage step, $V_{inc}$, is set to be $V_\pi/10$ for finding the approximate bias points and it can be changed.

In the second step of the algorithm, to search the precise bias points, $V'_{b3}$ found in the previous step is used and the voltage adjustment step, $V'_{inc}$, is set to be $V_\pi/100$ for accurate bias points search. While $V_{b3}$ is fixed to $V'_{b3}$, $V_{b1}$ and $V_{b2}$ are changed in the $2V_\pi$ range to find the bias condition $V_{b1}^{max}$ and $V_{b2}^{max}$, which make the intensity of the DPMZM output optical signal to be maximum. After select $V_{b1}^{max}$ and $V_{b2}^{max}$, $V_{b3}$ is swept in the $2V_\pi$ range and scanned the value which makes the output signal intensity of the DPMZM minimize, as expressed as $V_{b3}^{min}$. Therefore, the searched bias points, $V_{b1}^{max}$, $V_{b2}^{max}$, and $V_{b3}^{min}$ are the optimal bias points of the DPMZM for signal quadrupling because $V_{b1}^{max}$ and $V_{b2}^{max}$ make the two sub-MZMs operate at the MATP and the main MZM operates at the MITP if $V_{b3}^{min}$ is applied.

## 3. Validation of the Proposed Technique

### 3.1. Temperature Test

Since the bias drift phenomenon of the modulator is caused by temperature variation, the temperature test is conducted to verify the performance of the proposed automatic bias control technique. Figure 2 is the picture of the setup for the temperature test for verification of the proposed automatic bias control technique. In Figure 2, the automatic bias controller is in the temperature chamber with the DPMZM. For the experiment, the output power of the CW laser source (TeraXion Inc.: Quebec city, QC, Canada, PS-NLL-1550) is set to be 18 dBm at a 1550-nm wavelength. The AWG (Keysight: Santa Rosa, CA, USA, M8195A) generates signals of 2.5 GHz and 5.5 GHz single-frequency according to the test condition, respectively. The generated signal is fed to a hybrid coupler (Pasternack: Irvine, CA, USA, PE2058) and is split into two signals with a 90-degree phase difference. The split signals enter the RF ports of the DPMZM. The DPMZM (Fujitsu: Tokyo, Japan, FTM7962EP) has a half-wave voltage ($V_\pi$) and a 3-dB bandwidth of 3.5 V and 22 GHz, respectively. The output signal of the DPMZM is split into two signals through a tab coupler (AFR: Zhuhai, China, PMTC-55-2-10-N-B-P-1-F). One is fed to a tab PD (Kyosemi: Kyoto, Japan, KPDE008S) and the other is applied to a transmit PD (Finisar: California, CA, USA, XPD2120RA). The output signal of the tab PD enters a controller. The controller is composed of a commercial FPGA (field programmable gate array, Xilinx Inc.: San Jose, CA, USA, XC7CA75T-1CSG324I), DSP (digital signal processor, Texas Instruments Inc.: Dallas, TX, USA, TMS320C7748) chip, and other auxiliary electronic components such as flash memories (Micron Technology Inc.: Boise, ID, USA, MT25QL512ABB1EW9-0SIT), a power management IC (Texas Instruments Inc., LM26480SQ-AA/NOPB), an analog-to-digital converter (ADC, Analog Devices: Wilmington, MA, USA, AD7689ACPZ ), and an amplifier(Analog Devices, AD8659ACPZ). The output signal of the tab PD is used for monitoring the current status of the DPMZM, memorizing information such as output signal power, the bias values, the tendency of the bias voltages variation, and adjusting the bias voltages of the DPMZM. Therefore, the searched bias voltages are displayed on a monitor outside the chamber, and the output signal power of the DPMZM is measured by a spectrum analyzer during the temperature test.

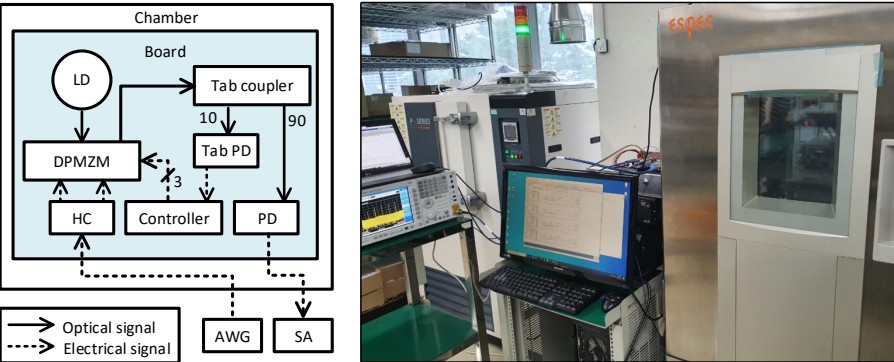

**Figure 2.** Temperature test set-up for verification of the proposed automatic bias control technique. AWG: Arbitrary waveform generator, DPMZM: Dual-parallel Mach–Zehnder modulator, PD: Photodetector, HC: Hybrid coupler, LD: Laser diode, and SA: Spectrum analyzer.

Figure 3 shows the temperature variation of the board which has the DPMZM and its automatic bias control module according to the temperature changes inside the chamber. In Figure 3, the blue dotted line represents the chamber temperature and the red solid line represents the board temperature. As shown in Figure 3, the temperature of the chamber increases from 20 °C to 50 °C at intervals of 3 °C. The temperature decreased to 30 °C at intervals of 5 °C. The rate of temperature change over time is 1.5 °C/min, and the temperature of the chamber is adjusted every 30 min so that the temperature of the board does not reach thermal equilibrium state. Since the temperature of the chamber and the temperature of the DPMZM control board are not in the thermal equilibrium state, the temperature of the board continuously changes, and the variation of the bias voltage can be continuously searched by using this nonequilibrium state. The proposed algorithm was done 600 times at a given temperature test condition and the bias voltages scan time was 45 s on average.

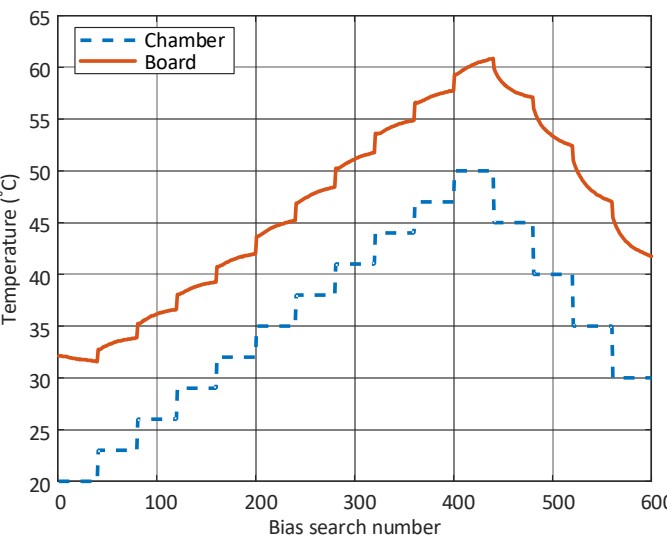

**Figure 3.** Temperature variations of the board according to the temperature changes inside the chamber.

### 3.2. Quadrupled Signal Generation

Figure 4 shows the searched bias voltages of the DPMZM according to temperature variation. In Figure 4, the temperature of the board is represented as the purple-solid line. The solid-blue-circle and the solid-red-square lines indicate the bias voltages of the two sub-MZMs, and the solid-yellow-diamond line denotes the bias voltage of the main MZM. As shown in Figure 4, the bias voltages for the generation of the quadrupled signal of the DPMZM change according to the temperature variation.

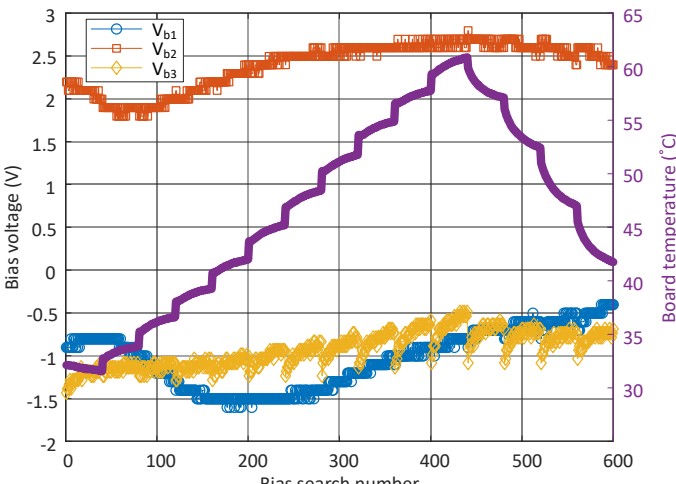

**Figure 4.** Searched bias voltages according to the temperature variation.

Figure 5 shows the intensity of the generated signals for each frequency. In Figure 5, the ordinal number means a signal with a frequency equal to a multiple of the frequency of the IF signal. As shown in Equation (1), the output signal of DPMZM includes signals having a frequency equal to a multiple of the IF signal frequency. As shown in Figure 5, it is noted that the intensity of the quadrupled signal generated by the proposed technique is constant and is 10 dB or more than the intensity of the adjacent signals. Therefore, it is verified that if the proposed technique is applied, the quadruple signal is generated regardless of temperature fluctuation.

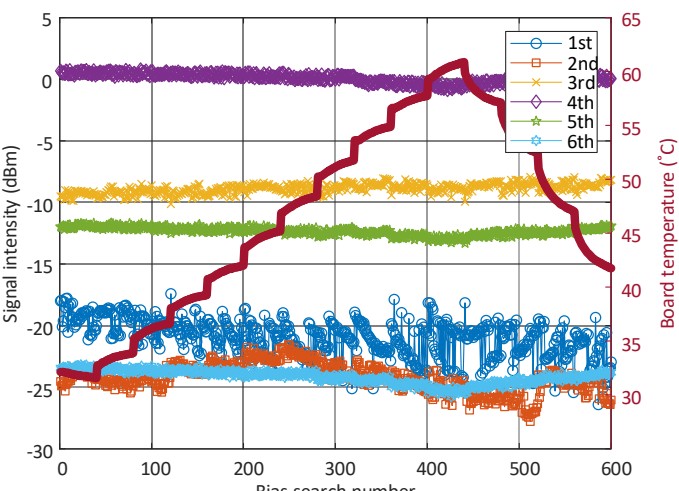

**Figure 5.** Intensity of the generated signals for each frequency.

Figure 6 shows the spectrum of the generated signal of the DPMZM by the proposed technique. Figure 6a,b show the generated signals with the frequency of 10 GHz and 22 GHz using IF signals with the frequency of 2.5 GHz and 5.5 GHz, respectively. As shown in Figure 6a,b, it is confirmed that the proposed technique can generate signals of X- and K-band signal according to the applied IF signal.

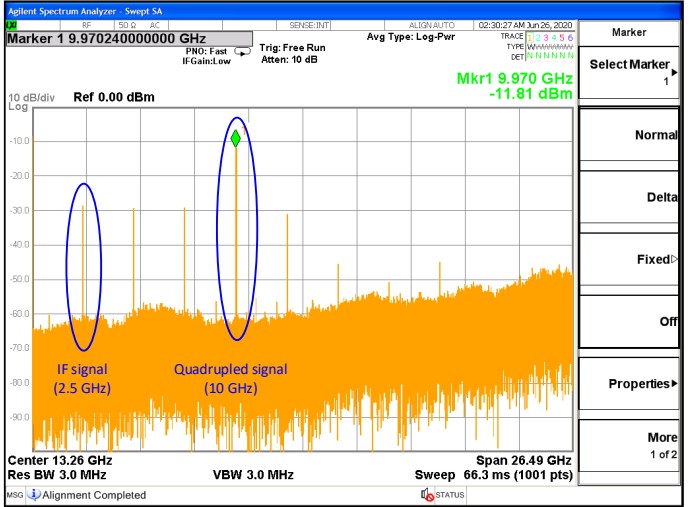

(**a**) 10 GHz signal generation.

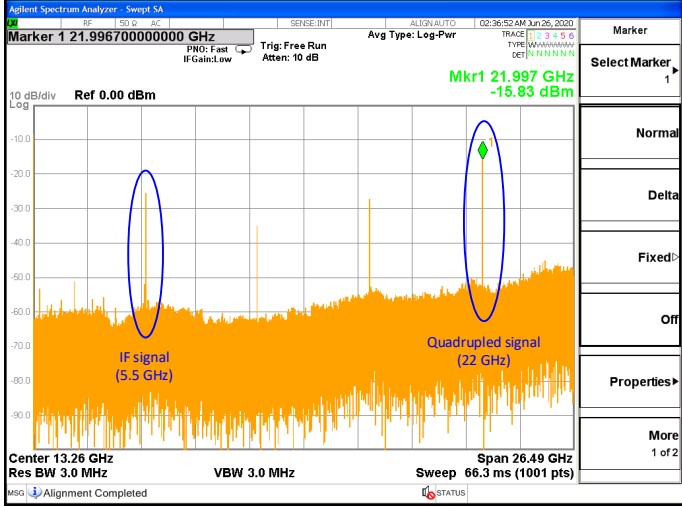

(**b**) 22 GHz signal generation.

**Figure 6.** The generated signal of the DPMZM by the proposed automatic bias technique. (**a**) The signal with a frequency of 10 GHz generated based on the intermediate frequency (IF) signal with a frequency of 2.5 GHz. (**b**) The signal with a frequency of 22 GHz generated based on the IF signal with a frequency of 5.5 GHz.

## 4. Discussion

Since the inherent bias drift phenomenon has a more complicated nature, stabilization of the searched bias conditions is required in a real operating environment [7]. The proposed automatic bias controller is tested in harsh environments than a real operating condition. Therefore, it can be verified that the proposed technique works properly in a real stable condition.

Table 1 shows the comparison with the previous research results of the automatic bias control techniques. As shown in Table 1, the method can be different according to the characteristics of the applied system. Most of the previous researches are performed for the application of optical communications, however, this work focuses on RF signal generation using the modulator. It is noted that this work conducts the longest verification test duration with the harsh temperature variation.

**Table 1.** Comparison with the previous research results of automatic bias control techniques.

|  | [10] | [11] | [12] | [13] | [14] | [15] | [16] | **This work** |
|---|---|---|---|---|---|---|---|---|
| Methods | dithering | power monitoring | dithering | power monitoring | dithering | power monitoring | dithering | **power monitoring** |
| Verification | time (6.5 h) | eye diagram | time (35 min) | eye diagram | time (35 min) | BER * measurement | time (40 min) | **temperature (11 h)** |
| Applications | QPSK | QPSK | QPSK | - | ADC | QPSK | QPSK | **RF signal generation** |

* BER: Bit error rate; QPSK: Quadrature phase shift keying; ADC: Analog-to-digital converter; RF: Radio frequency.

The temperature test results of the proposed automatic bias search technique are shown in Figures 4 and 5. During the temperature test, the board in the chamber is not in the thermal equilibrium state, therefore, the temperature of the board changes continuously. Since the temperature of the board does not settle, the proposed algorithm should execute during the whole temperature test period. The proposed algorithm has been performed 600 times during the temperature test and the generation of the quadrupled signal conducted without any failure.

## 5. Conclusions

In this paper, we proposed an automatic bias control technique based on a simulated annealing algorithm for generating the quadrupled signal. The proposed technique sepa-

rates the bias condition search of the two sub-MZMs and main MZM, and combines the local and global optimum bias conditions. The performance of the proposed technique was verified through a temperature test. A total number of 600 bias searches were performed while the temperature of the chamber changed, and the quadrupled signal was successfully generated in real-time in all temperature test ranges. In addition, it was verified that the proposed technique generated signals with a center frequency of 10 GHz and 22 GHz using IF signals with a center frequency of 2.5 GHz and 5.5 GHz, respectively.

Therefore, the proposed automatic bias control technique of the DPMZM is promising for the various applications requiring frequency up-conversion using microwave photonics technology.

**Author Contributions:** Conceptualization, S.J. and Y.B.; methodology, M.Y. and S.Y.; validation, Y.B. and M.Y.; formal analysis, Y.B.; investigation, J.S.; resources, J.R.; data curation, S.Y. and Y.B.; writing—original draft preparation, Y.B.; writing—review and editing, redY.B.; visualization, Y.B.; supervision, J.S. and S.J.; project administration, J.S.; funding acquisition, J.S. All authors have read and agreed to the published version of the manuscript.

**Funding:** This research received no external funding.

**Institutional Review Board Statement:** Not applicable.

**Informed Consent Statement:** Not applicable.

**Data Availability Statement:** Not applicable.

**Conflicts of Interest:** The authors declare no conflict of interest.

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
