# Peer review of "Automatic Bias Control Technique of Dual-Parallel Mach–Zehnder Modulator Based on Simulated Annealing Algorithm for Quadrupled Signal Generation"

_photonics, doi:10.3390/photonics8030080_

Round 1

Reviewer 1 Report

The paper is devoted to the problem of searching and stabilization of operating bias voltages of a dual-parallel Mach-Zehnder modulator (DPMZM). A special algorithm for the optimal bias voltages search was developed and tested for the varying temperature. The paper has some merit for DPMZM applications but must be significantly improved for the problem comprehension. My comments are here.
1. The introduction needs to be improved. Authors told that previous work deal mostly with QPSK modulation. However specific features and problems related to the particular application of quadrupled signal generation were not exposed.
2. More detail description of the experimental realization is needed. What optical and electronic components were used for bias control and them required characteristics? What components comprised feedback system?
3. The algorithm of the optimal bias voltages search is given but what algorithm was used for the operating point stabilization? Perform complicated algorithm for the stabilization very inconvenient, slow and could lead to the pauses in the quadrupled signal generation.
4. Note that stabilization is required not only for compensation of the temperature variation. Inherent DC drift has more complicated nature thus bias control should be used even at stable environmental conditions.

Reviewer 2 Report

The paper describes an automated bias control technique applied to a well known scheme for the generation of fourth harmonic of the input RF signals. The description of the scheme exploited could be shortened considerably since all the relevant info is clearly introduced e.g. in Ref. 15 of the paper. The scheme used exploits two MZM biased in the ON state plus one biased in the OFF state; the purpose of the temperature control loop seems to work directly on the detection of the fourth harmonic. From this standpoint the experimental setup should be better described:

  • what are the characteristics of the MZM exploited? it is difficult to appreciate the details of the experiment without knowing e.g. what are the V_pi of the modulators.
  • same for details on the laser source, PD etc.
  • what is the feedback delay time (i.e. the time required to the system to readjust the bias after a temperature step?
  • Any comment on how this setup could be exploited in realistic conditions, i.e. when the output signal has to be monitored but not completely exploited for control?

The Introduction includes some wrong statements:

  • "The relation between the output optical signal and the applied driving voltage has nonlinear properties because of anisotropic dielectric characteristics of the materials". This is completely wrong (it was also wrong in Ref. 6 from which the sentence is taken), only LiNbO3 is anisotropic (uniaxial) while InP and GaAs are isotropic. The "nonlinear properties" derive from the electro-optic effect whereby (in those materials) the refractive index changes with the applied field (mainly in a linear way, according to the Pockels effect).
  • The bias point instability is, from a physical standpoint, quite a complex story. Most of the effects referred to actually are specific to LiNbO3 rather than to other MZM technologies, and, besides, temperature variations are not the only culprits. I would confine the text to citing papers where the issue of stabilization is discussed, without going into details or citing almost verbatim piece of sentences from other sources. After all the technique proposed is OK independent of the cause of bias migration.

Round 2

Reviewer 1 Report

Authors was not answered my previous comments completely.

  1. The introduction needs to be improved. Authors told that previous work deal mostly with QPSK modulation. However specific features and problems related to the particular application of quadrupled signal generation were not exposed.

Since the phase bias of the DPMZM should be set to be the different point compared with the optical QPSK modulation, a different bias search technique is required.

It is very formal answer. What working point is required? Is it depends on the modulation signal? What accuracy and stability are required? What rate for bias voltage change is needed to compensate the bias voltage drift? The question like these should be answed.

  1. More detail description of the experimental realization is needed. What optical and electronic components were used for bias control and them required characteristics? What components comprised feedback system?

Authors gave brief description of the experimental setup but have nod disclose working principles of the bias voltage controller which is the object of investigations.

  1. The algorithm of the optimal bias voltages search is given but what algorithm was used for the operating point stabilization? Perform complicated algorithm for the stabilization very inconvenient, slow and could lead to the pauses in the quadrupled signal generation.

The proposed algorithm was done 600 times at a given temperature test condition and the bias voltages scan time was 45 seconds on average.

What happened with quadrupled signal generation in the process of the bias voltage scanning when bias voltage took not optima values?

  1. Note that stabilization is required not only for compensation of the temperature variation. Inherent DC drift has more complicated nature thus bias control should be used even at stable environmental conditions.

In the revised manuscript I have not found response on this comment at all.

The last comment is related terminology. The right term is refractive index rather than reflective index. Also nonlinearity is not caused by electrooptic effect which is linear effect. Nonlinear transfer function is result of the light interference.

Reviewer 2 Report

no further comments.

Author Response

We sincerely appreciate for review of the manuscript. The manuscript has been improved by the reviewers' valuable comments.

Round 3

Reviewer 1 Report

Authors tried to answer all my comment. The last version of the manuscript is not ideal Nevertheless I am recommended it for publication.